# Cerebrospinal Fluid and Blood Biomarkers in Patients with Post-Traumatic Disorders of Consciousness: A Scoping Review

**DOI:** 10.3390/brainsci13020364

**Published:** 2023-02-20

**Authors:** Sergio Bagnato, Cristina Boccagni

**Affiliations:** Unit of Neurophysiology and Unit for Severe Acquired Brain Injuries, Rehabilitation Department, Giuseppe Giglio Foundation, viale G. Giardina, 90015 Cefalù, PA, Italy

**Keywords:** traumatic brain injury, vegetative state, unresponsive wakefulness syndrome, minimally conscious state, diagnosis, prognosis, plasticity, neurodegeneration, biomarker, neurofilament light chain

## Abstract

(1) Background: Cerebrospinal fluid (CSF) and blood biomarkers are emerging tools used to obtain information on secondary brain damage and to improve diagnostic and prognostic accuracy for patients with prolonged post-traumatic disorders of consciousness (DoC). We synthesized available data from studies evaluating CSF and blood biomarkers in these patients. (2) Methods: A scoping review was conducted according to the Preferred Reporting Items for Systematic Reviews and Meta-Analyses extension for Scoping Reviews checklist to identify and synthesize data from relevant studies. Studies were identified by PubMed and manual searches. Those involving patients with unresponsive wakefulness syndrome or in a minimally conscious state for >28 days, evaluating CSF or blood biomarkers, and conducted on patients with traumatic brain injuries older than 16 years were included in the review. (3) Results: In total, 17 studies were included. Findings on neurofilament light chain, proteins, metabolites, lipids, amyloid-β, tau, melatonin, thyroid hormones, microtubule-associated protein 2, neuron-specific enolase, and brain-derived neurotrophic factor were included in the qualitative synthesis. (4) Conclusions: The most promising applications for CSF and blood biomarkers are the monitoring of secondary neurodegeneration, support of DoC diagnoses, and refinement of prognoses, although current evidence remains too scarce to recommend such uses of these biomarkers in clinical practice.

## 1. Introduction

Severe traumatic brain injury (TBI) poses a worldwide public health challenge because of its high mortality and residual disability rates [1]. After the acute phase, many patients with severe TBI evolve from coma to other disorders of consciousness (DoC), the duration of which is often difficult to forecast and is defined as prolonged when lasting more than 4 weeks [2]. The recognition of a prolonged DoC following coma is currently based on patients’ behavioral characteristics. Unresponsive wakefulness syndrome (UWS), also known as the vegetative state, is characterized by a complete lack of self- and external-world awareness—like coma—but differs from coma in that patients show spontaneous eye-opening and -closing cycles [3]. The minimally conscious state (MCS), which may follow coma or UWS, is identified by the reappearance of different degrees of awareness, although patients remain incapable of functional communication or object use [4]. 

Cerebrospinal fluid (CSF) and blood biomarkers have been proposed for brain injury assessment and outcome prediction in the acute phase of severe TBI. Among the most thoroughly studied acute-phase biomarkers, S100B [5,6], ubiquitin C-terminal hydrolase-L1 [7], glial fibrillary acid protein [8,9], neuron-specific enolase [10,11], tau protein [12,13], and neurofilament light chain [14] levels correlate with outcomes evaluated at different post-injury timepoints. CSF and blood biomarkers of brain injury have been less well studied in patients with prolonged post-traumatic DoC, but they potentially provide useful information in this context. Their most obvious application is for a better characterization of the extent of brain injury in the post-acute phase, which would provide information to be integrated in prognostic models. As this field of research is relatively new, current guidelines for DoC prognosis emphasize the roles of standardized behavioral evaluations and neurophysiological and neuroimaging assessments, with no mention of CSF or blood biomarkers [15]. Moreover, these biomarkers may help clinicians to better understand the mechanisms of secondary brain injury after TBI. Severe TBI may trigger prolonged neurodegeneration, increasing the risk of several neurodegenerative diseases, including Alzheimer’s and Parkinson’s diseases [16,17]. In patients with prolonged DoC, this secondary neurodegeneration may further affect the chance of recovering consciousness and worsen functional outcomes. In this context, biomarkers used to evaluate damage to different cell populations (e.g., neurons or glial cells) or in different parts of the same cells (e.g., cell body or axon) provide the opportunity to better characterize ongoing brain degeneration, as such information cannot be obtained at the same level of detail with neuroimaging or neurophysiological techniques.

We performed a scoping review of studies evaluating the application of CSF and blood biomarkers to the evaluation of prolonged post-traumatic DoC. Its results might be useful for clinicians and researchers interested in planning studies in this young field of research.

## 2. Methods

### 2.1. Search Strategy 

A search of the PubMed/Medline electronic database was performed on 15 December 2022, according to the Preferred Reporting Items for Systematic Reviews and Meta-Analyses extension for Scoping Reviews checklist [18]. The following terms, combined via Boolean logical operators to reduce the number of non-pertinent results, were used: (“biomarker” OR “biomarkers”) AND (“plasma” OR “serum” OR “blood” OR “cerebrospinal fluid”) AND (“disorders of consciousness” OR “vegetative state” OR “unresponsive wakefulness syndrome” OR “minimally conscious state”). To find additional suitable material, the reference sections of selected articles were searched manually and the authors’ personal databases on the topic were searched.

### 2.2. Eligibility Criteria and Data Extraction

Potentially eligible studies obtained from the systematic search were identified by the screening of titles and abstracts and then assessed via full-text review. Studies that (1) involved clinical research, (2) involved the evaluation of CSF or blood biomarkers in subjects with prolonged (i.e., >28 days) DoC following coma, (3) were conducted with subjects with TBIs older than 16 years, and (4) were published in English, were included in the review. Both authors performed the screening and full-text review, achieving consensus by brief discussion in case of disagreement at any stage. The data extracted from each study were the authors’ names and year of publication, study design, participant characteristics [number, age, DoC, etiology (exclusively or non-exclusively TBI), and time since brain injury], biomarkers evaluated, and main findings. These data were qualitatively synthesized. An overview of the search and review procedures is provided in Figure 1.

## 3. Results

### 3.1. Study Selection and Characteristics

The systematic and manual searches generated 60 records. After title and abstract screening, the full texts of 17 articles were reviewed for the determination of eligibility. All 17 articles were included in the review (Table 1) [19,20,21,22,23,24,25,26,27,28,29,30,31,32,33,34,35]. 

All studies had observational designs, with data collected retrospectively in 11 [21,22,24,25,26,27,28,30,31,32,35] and prospectively in 6 [19,20,23,29,33,34] studies. The numbers of patients with post-traumatic DoC ranged from 3 to 52. All but two [29,33] studies included patients with UWS and MCS. The results are reported below according to the potential biomarkers evaluated.

### 3.2. Neurofilament Light Chain

Neurofilament light chain levels were evaluated in four studies [19,20,21,22], one of which also involved the evaluation of the serum glial fibrillary acidic protein level (not reported on in this review because it was assessed in only three patients) [22]. Neurofilament light chain levels were analyzed in serum in three studies [19,20,22] and in CSF in one study [21]. Two studies were conducted only with patients with TBI [19,21], and one study each was conducted with patients with TBI and hypoxic-ischemic brain injury [20] and DoC of mixed etiologies [22]. All studies included patients with UWS and those in MCS. 

Neurofilament light chain levels were higher in patients with prolonged DoC than in healthy controls [19,20,22] and patients with Alzheimer’s disease [21], a population characterized by high neurofilament light chain levels due to neuronal degeneration. The magnitudes of neurofilament light chain increase in patients with post-traumatic DoC ranged from 2.4- to 60.5-fold the upper normal limit in CSF [21] and were 13.4- and 3.7-fold those in healthy subjects in serum at 1–3 and 6 months post-injury, respectively [20]. Serum neurofilament light chain levels were higher in patients with UWS than in those in a MCS at 1–3 months post-injury, and lower in patients with TBI than in those with hypoxic-ischemic brain injury at 6 months post-injury [20]. Neurofilament light chain levels did not differ between patients with hemorrhagic and non-hemorrhagic TBIs up to 6 months post-injury [19]. Levels in blood and CSF decreased over time in all studies [19,20,21,22].

### 3.3. Proteins, Metabolites, and Lipids

This heterogeneous group of biomarkers was evaluated in four studies [23,24,25,26], in blood in three studies [23,25,26], and CSF in one study [24]. All studies included patients with UWS and MCS. In one study, differences in plasma protein expression between patients with post-traumatic DoC and healthy subjects were assessed using a proteomics approach; of 300 proteins containing at least one identified peptide, 32 were differentially expressed [23]. The main abnormalities were found in the complement cascade, with upregulation involving mainly C-reactive protein, complement components C3 and C9, complement component C8 alpha chain, and mannose-binding protein C [23].

In another study, CSF protein levels in patients with DoC of mixed etiologies selected for spinal cord stimulation were analyzed retrospectively [24]. After spinal cord stimulation, consciousness improved in 12 patients (5 with TBIs) and did not improve in 54 patients (22 with TBIs), with CSF protein levels in the latter group more than double those in the former (101 vs. 45 mg/L) [24]. In another retrospective study conducted with patients with prolonged DoC of mixed etiologies, low serum albumin and hemoglobin levels and high white blood cell counts correlated with mortality [25].

Metabolomics analyses revealed 21 metabolic pathways that were down- or upregulated in patients with post-traumatic DoC, of which the purine metabolism pathway was most affected, with significant decreases in adenosine triphosphate degradation products (adenosine, adenosine monophosphate, and adenosine diphosphate) [26]. The glycine, serine, and threonine metabolism pathways were the most affected among upregulated pathways [26]. Lipidomics analyses revealed 20 classes of lipids differentially expressed (primarily reduced) in patients with DoC [26]. Notably, phosphatidylcholine and arachidonic acid levels differed substantially between patients with UWS and those in MCS [26].

### 3.4. Amyloid-β and Tau Proteins

CSF amyloid-β levels were evaluated in two studies [27,28], one of which also involved the analysis of total and phosphorylated tau levels [27]. Both studies were conducted exclusively with patients with DoC caused by TBIs. Reduced amyloid-β levels were found in 87.5–93.3% of patients, with no difference between patients with UWS and those in MCS and no correlation with the time since brain injury [27,28]. Total tau levels were normal in all patients, whereas phosphorylated tau levels were increased slightly in 1 of 15 patients [27].

### 3.5. Melatonin and Thyroid Hormones

Blood hormone levels were evaluated in two studies [29,30]. Melatonin was studied in a small sample of patients with post-traumatic UWS; its level did not increase over time and was not suppressed by light relative to that in healthy subjects [29]. The thyrotropic axis [thyroid-stimulating hormone (TSH), free triiodothyronine, and free thyroxine (fT4)] was evaluated at admission and after 6 months of rehabilitation in a study conducted with patients with DoC of mixed etiologies [30]. Lower baseline TSH levels and greater TSH increments after rehabilitation were associated with better outcomes, independently of the DoC etiology and type [30]. Moreover, smaller fT4 changes over time were associated with better functional outcomes and cognitive function [30].

### 3.6. Microtubule-Associated Protein 2

Microtubule-associated protein 2 was evaluated in a study conducted with patients with post-traumatic DoC and those with higher levels of consciousness [31]. At 6 months post-injury, microtubule-associated protein 2 levels were higher in patients than in controls and in patients with UWS than in those with higher levels of consciousness [31].

### 3.7. Neuron-Specific Enolase

Neuron-specific enolase was evaluated in one study [32]. Its level was lower in patients than in heathy controls, due mainly to lower levels in patients at ≥12 months than <12 months after TBI [32].

### 3.8. Brain-Derived Neurotrophic Factor

Brain-derived neurotrophic factor was evaluated in one study conducted with patients with DoC of mixed etiologies [33]. Its level was lower in patients than in healthy controls and was not increased by verticalization with robot-assisted lower-limb training [33].

### 3.9. Soluble Neural Cell Adhesion Molecule

Soluble neural cell adhesion molecule was assessed before, during, and after 2 weeks of anodal transcranial direct-current stimulation in patients in MCS [34]. Post-treatment outcomes were better in patients with low levels of this molecule [34].

### 3.10. MicroRNAs

MicroRNAs were studied in a small sample of patients with post-traumatic DoC [35]. The expression of 41 microRNAs differed between patients and controls; 30 were upregulated and 18 were downregulated [35].

## 4. Discussion

This scoping review explored the use of CSF and blood biomarkers in patients with post-traumatic DoC. Three main potential fields of application for these biomarkers emerged: the monitoring of ongoing secondary brain injury and recovery mechanisms, the support of diagnoses among DoC, and the refinement of prognostic judgments.

### 4.1. Monitoring of Ongoing Secondary Brain Injury and Recovery Mechanisms

This application is particularly suitable for biomarkers that directly or indirectly reflect neuronal damage, neurodegeneration, or brain plasticity involved in recovery processes. Among the biomarkers examined in the studies included in this review, neurofilament light chain, microtubule-associated protein 2, neuron-specific enolase, amyloid-β, tau, and brain-derived neurotrophic factor belong to this category. Severe TBI may trigger prolonged neurodegeneration via mechanisms that remain incompletely recognized but probably involve chronic inflammation with progressive white- and gray-matter atrophy [36,37]. Recovery from post-traumatic DoC requires the activation of several brain plasticity mechanisms to reconstitute neural connections and pathways [38]. Levels of neurofilament light chain and microtubule-associated protein 2 are increased for up to 6 months in blood [19,20,22,31] and up to 19 months in CSF [21] after TBI, potentially reflecting prolonged neurodegeneration affecting axons and dendrites, respectively [39,40]. The level of neuron-specific enolase, which likely reflects cell body damage [41], was found to be reduced at >1 year after TBI; although this finding cannot be interpreted definitively, it may reflect reduced release due to greater brain atrophy in patients with longer DoC durations [32].

TBI increases the risk of amyloid-β–related pathologies, such as Alzheimer’s disease [42]. Reduced CSF amyloid-β levels may reflect reduced clearance and deposition in the brains of patients with post-traumatic DoC [27,28]. In one study, however, abnormal amyloid-β metabolism was not accompanied by signs of tau pathology [27], which is required for the development of Alzheimer’s disease.

Neurotrophic factors play a critical role in regulating neuronal differentiation and in neuronal survival, growth, and plasticity both in normal and pathological conditions. Multiple neurotrophin–receptor interactions have been reported in the brains of mammals after TBI [43]. Only one study included in this review involved the examination of a biomarker involved in brain plasticity [33]. It showed that the serum level of brain-derived neurotrophic factor, a neurotrophin involved in neurogenesis and synaptic plasticity [44], is reduced in patients with prolonged DoC, potentially reflecting increased use of circulating brain-derived neurotrophic factor by the brain [33]. This level was not increased by verticalization with robot-assisted lower-limb training [33]; nevertheless, the ability of other rehabilitative approaches to stimulate brain plasticity in these patients should be assessed.

### 4.2. Support of Diagnoses among DoC

The diagnosis of prolonged DoC is currently based on multimodal evaluation, which may integrate standardized neurobehavioral assessments with electroencephalogram-based and functional neuroimaging techniques [45]. Biomarkers used to support different diagnoses must distinguish between patients with UWS and those in MCS. Findings of studies examined in this review include higher neurofilament light chain levels at 3 months after TBI in patients with UWS than in those in the MCS [20], and lower microtubule-associated protein 2 levels in patients with UWS than in conscious patients (including those in MCS and higher levels of consciousness) [31]. However, these data provide little help for differential diagnosis at the individual patient level. The combined evaluation of lipids [phosphatidylcholine (38:5)-H and arachidonic acid] seems to enable the distinction of UWS and MCS in individual patients with adequate accuracy, although further confirmation of this finding is required [26].

### 4.3. Refinement of Prognostic Judgments

The accurate determination of DoC prognosis is very challenging, with obvious implications in terms of appropriateness of specific rehabilitative treatments and resource allocation. This review identified some routine tests with potential usefulness for this purpose. Lower baseline TSH levels and higher TSH increments after 6 months were associates with better functional and cognitive outcomes in patients with prolonged DoC, although the underlying mechanisms remain to be elucidated [30]. Biomarkers related to the nutritional state (albumin and hemoglobin) or the presence of infection (white blood cell count) are predictors of death in this patient group [25]. Finally, the CSF protein level may be useful for the identification of patients who will benefit from spinal cord stimulation [24], and the soluble neural cell adhesion molecule level is related to outcomes after transcranial direct current stimulation [34].

### 4.4. Study Limitations and Future Research Directions

The main limitation of this scoping review is the low level of evidence of the included studies. No randomized controlled trial was identified by the systematic search; all studies were observational, and most were retrospective. Some studies included patients with DoC of mixed etiologies, with no separate analysis of data from those with post-traumatic DoC. Moreover, the numbers of patients with post-traumatic DoC included in some studies were very small. Finally, our search may have missed some relevant studies as it was limited to the PubMed/Medline database.

At the time of this scoping review, the applications discussed above are the most promising for CSF and blood biomarkers in patients with post-traumatic DoC, although further evidence is needed to justify their use in routine clinical practice. The simultaneous examination of multiple biomarkers could improve the ability to monitor secondary neurodegeneration and improve diagnostic and prognostic accuracy. To achieve this goal, additional well-designed studies, especially randomized controlled trials, are needed.

## Figures and Tables

**Figure 1 brainsci-13-00364-f001:**
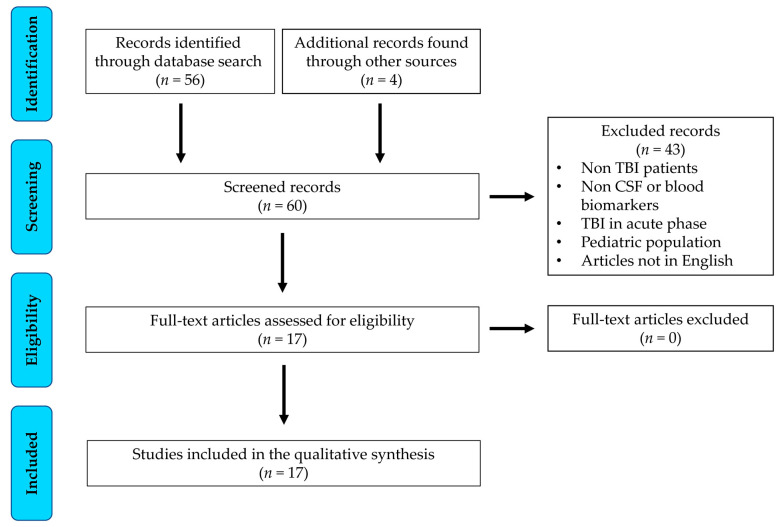
Flow of the scoping review.

**Table 1 brainsci-13-00364-t001:** Summary of studies evaluating CSF and blood biomarkers in patients with prolonged DoC.

Reference	Biomarker	Fluid	Population (*n*)	Etiology (*n*)	Time Post-Injury at Study Entry	Main Findings
Bagnato et al., 2022 [19]	NFL	Serum	UWS (25) MCS (27) HC (52)	Traumatic (52)	28–90 days	NFL level higher in patients vs. controls at baseline and 6 months post-injury, no difference according to intracranial hematoma
Bagnato et al., 2021 [20]	NFL	Serum	UWS (45) MCS (25) HC (70)	Traumatic (48) Hypoxic (22)	28–90 days	NFL levels higher in patients vs. controls at baseline and 6 months post-injury, patients with UWS vs. those in MCS at baseline, patients with hypoxic brain injury vs. those with TBI 6 months post-injury
Bagnato et al., 2017 [21]	NFL	CSF	UWS (3) MCS (7) AD (9)	Traumatic (10)	95–581 days	NFL level higher in patients vs. normal limit and patients with AD
Coppola et al., 2022 [22]	NFL and GFAP	Serum	UWS (7) MCS (9) HC (6)	Traumatic (3) Hypoxic (6) Vascular (7)	1–14 months	NFL and GFAP levels higher in patients vs. controls
Bao et al., 2018 [23]	300 different proteins	Plasma	UWS (13) MCS (5) HC (6)	Traumatic (18)	30–254 days	32 proteins, especially those involved in complement cascade, differentially expressed in patients vs. controls
He et al., 2022 [24]	Proteins	CSF	UWS (24) MCS (42)	Traumatic (27) Hypoxic (11) Vascular (28)	3–>12 months	High protein levels associated with poor outcomes after spinal cord stimulation
Romaniello et al., 2016 [25]	Albumin, hemoglobin, white blood cells count	Serum, whole blood	UWS (N/A) MCS (N/A)	Traumatic (25) Hypoxic (42) Vascular (40) Other (5)	11 months (average)	Albumin and hemoglobin levels, white blood cell count correlate with mortality
Yu et al., 2021 [26]	Metabolomic and lipidomic profiles	Plasma	Metabolomic profile - UWS (12) - MCS (11) - EMCS (15) - AD (19) - HC (8) Lipidomic profile - UWS (32) - MCS (22) - HC (32)	Traumatic (92; some patients studied both in UWS and MCS)	N/A	Purine metabolism pathway suppressed in patients with UWS and MCS; some lipids distinguish these patients
Bagnato et al., 2018 [27]	Amyloid-β, total tau, phosphorylated tau	CSF	UWS (3) MCS (12)	Traumatic (15)	92–578 days	Amyloid-β level reduced
Bagnato et al., 2017 [28]	Amyloid-β	CSF	UWS (1) MCS (7)	Traumatic (8)	95–578 days	Amyloid-β level reduced
Guaraldi et al., 2014 [29]	Melatonin	Plasma	UWS (6) HC (9)	TBI (6)	6–18 months	Melatonin synthesis reduced at night, not suppressed by light
Mele et al., 2022 [30]	TSH, fT3, fT4	Serum	UWS (94) MCS (57)	Traumatic (45) Hypoxic (33) Vascular (73)	28–90 days	Lower baseline TSH level, greater TSH increment after rehabilitation predict good outcomes
Mondello et al., 2012 [31]	MAP-2	Serum	UWS (5) MCS (4) EMCS (7) HC (16)	Traumatic (16)	6 months	MAP-2 level lower in patients with UWS vs. those with higher levels of consciousness
Bagnato et al., 2020 [32]	NSE	Serum	UWS (14) MCS (21) EMCS (16) HC (30)	Traumatic (51)	23 months (average)	NSE level lower at longer intervals after TBI
Bagnato et al., 2020 [33]	BDNF	Serum	UWS (10) MCS (8) HC 16)	Traumatic (8) Hypoxic (6) Vascular (4)	1–7 months	BDNF level reduced in patients, not modified by verticalization with robot-assisted lower-limb training
Ziliotto et al., 2019 [34]	sNCAM	Plasma	MCS (8) HC (39)	Traumatic (8)	1–19 years	Low sNCAM level associated with better outcomes after transcranial direct current stimulation
Zilliox et al., 2022 [35]	miRNAs	Whole blood	UWS (2) MCS (4)	Traumatic (6)	399–730 days	41 miRNAs differentially expressed in patients vs. controls

AD, Alzheimer’s disease; BDNF, brain-derived neurotrophic factor; CSF, cerebrospinal fluid; EMCS, emergence from minimally conscious state; fT3, free triiodothyronine; fT4, free thyroxine; GFAP, glial fibrillary acidic protein; HC, healthy control; MAP-2, microtubule-associated protein 2; MCS, minimally conscious state; miRNA, microRNA; N/A, not available; NFL, neurofilament light chain; NSE, neuron-specific enolase; sNCAM, soluble neural cell adhesion molecule; TBI, traumatic brain injury; TSH, thyroid-stimulating hormone; UWS, unresponsive wakefulness syndrome.

## Data Availability

The datasets generated during the current study are available from the corresponding author on reasonable request.

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
