# Peer review of "Cerebrospinal Fluid and Blood Biomarkers in Patients with Post-Traumatic Disorders of Consciousness: A Scoping Review"

_brainsci, 2023, doi:10.3390/brainsci13020364_

Round 1

Reviewer 1 Report

The manuscript presented from Bagnato et al., is interesting and original. However the authors should improve different critical points:

- The authors should improve the introduction and text concerning the role of neurotrophic factors after TBI. (For a review see, Cacialli P., International journal of molecular sciences 2021)

- The authors should improve the discussion concerning the limitation of their approach.

Author Response

The manuscript presented from Bagnato et al., is interesting and original. However the authors should improve different critical points:

- The authors should improve the introduction and text concerning the role of neurotrophic factors after TBI. (For a review see, Cacialli P., International journal of molecular sciences 2021).

Response: Thank you for your suggestion. We have improved the discussion on the role of neurotrophic factors in section 4.1 (Monitoring of ongoing secondary brain injury and recovery mechanisms). As per your recommendation, the text now reads: "Neurotrophic factors play a critical role in regulating neuronal differentiation and in neuronal survival, growth, and plasticity both in normal and pathological conditions. Multiple neurotrophin-receptor interactions have been reported in the brains of mammals after TBI [43]" (lines 219-223). Furthermore, we have added Dr. Ciacalli's paper in our reference list.

- The authors should improve the discussion concerning the limitation of their approach.

Response: We have strengthened the discussion of the limitations of our approach by including the statement that "our search may have missed some relevant studies as it was limited to the Pubmed/Medline database" (lines 262-263). This highlights the potential limitations of our findings and methodology.

Reviewer 2 Report

Drs. Bagnato and Boccagni provided a scoping review of studies assessing CSF and blood biomarkers in post-traumatic disorders of consciousness. The review demostrates that studies in this area are rather heterogenous in terms of patient populations and studied biomarkers, and poorly controlled. These factors preclude authors from providing sound recommendations on future research directions in this area (e.g., which biomarker to suggest when planning a study) or on selectinng tests for routine care. However, this paper gives an important overview of current situation and may stimulate further research efforts.

Considering limitations of the study, it might be useful to widen the search and to include papers indexed not only in PubMed/Medline. 

Author Response

Drs. Bagnato and Boccagni provided a scoping review of studies assessing CSF and blood biomarkers in post-traumatic disorders of consciousness. The review demostrates that studies in this area are rather heterogenous in terms of patient populations and studied biomarkers, and poorly controlled. These factors preclude authors from providing sound recommendations on future research directions in this area (e.g., which biomarker to suggest when planning a study) or on selectinng tests for routine care. However, this paper gives an important overview of current situation and may stimulate further research efforts.

Considering limitations of the study, it might be useful to widen the search and to include papers indexed not only in PubMed/Medline.

Response. We acknowledge the Reviewer's suggestion regarding the search of additional databases. Although we agree that it could have added further value to our study, we regret informing that this methodological aspect cannot be revised at this stage. However, we have strengthened the discussion of the limitations of our approach by including the statement that "our search may have missed some relevant studies as it was limited to the Pubmed/Medline database" (lines 262-263). This highlights the potential limitations of our findings and methodology.

Round 2

Reviewer 1 Report

The authors improved the manuscript.